# Invasive and Non-Invasive Fungal Rhinosinusitis—A Review and Update of the Evidence

**DOI:** 10.3390/medicina55070319

**Published:** 2019-06-28

**Authors:** Peter George Deutsch, Joshua Whittaker, Shashi Prasad

**Affiliations:** 1Department of Otolaryngology, University Hospital Coventry and Warwickshire, Coventry, CV2 2DX, UK; 2Department of Otolaryngology, Queens Medical Centre, Nottingham, NG7 2UH, UK

**Keywords:** fungal sinusitis, sinusitis, mucormycosis, allergy, rhinosinusitis, fungal, otolaryngology

## Abstract

Fungal infections are a subset of infectious processes that an otolaryngologist is required to be familiar with. They can be encountered in otology, rhinology and head and neck surgery. The presence of fungal rhinosinusitis is well recognised by otolaryngologists, but the classifications and appropriate management are not so well understood. The prevalence of fungal sinus disease is thought to be have been increasing in recent decades There is speculation that this may be due to increased awareness, antibiotic overuse and increased use of immunosuppressant medications. Added to this, there has been a large amount published on the role of fungi as a causative organism in chronic rhinosinusitis. Given the importance of fungal rhinosinusitis in clinical practice, we aim to review the classification and current management strategies based on up-to-date literature.

## 1. Introduction

### 1.1. Background

Fungal infections are one of the four major microbiological sub-groups. Fungi can take two different forms. These are yeasts, which are unicellular organisms, and moulds, which are branching filamentous organisms and hence are more easily identified. To some degree, fungi can go between these two states [1,2]. As with bacteria, viruses and parasites, there are many thousand different types of fungi. A much smaller number are commonly clinically relevant micro-organisms (20–25) [2,3]. The most commonly encountered fungal species in medical practice are *Candida* species and *Aspergillus* species [3]. The less commonly encountered, but known for their invasive potential, are fungi of the *Zygomycota* order (*Mucor*, *Rhizopus*, *Apophysomyces* et al.). These fungi are often implicated in immunocompromised individuals, as in the case of Mucormycosis [4]. Fungal spores are abundant in the atmosphere and so readily encounter anatomical structures relevant to ENT surgeons. These fungi, however, only develop pathological potential if the environment is suitable for this. In normal conditions, fungi that are inhaled form part of the normal sinonasal flora. These fungi are then destroyed by normal functioning immunological cascades. However, following the prolonged use of antibiotics, poor ventilation, dark and moist environments as well as immunocompromise, these immunological pathways are disrupted, making fungal infections more likely [3]. 

### 1.2. Fungal Rhinosinusitis

Fungal infections of the paranasal sinuses are in fact a spectrum of diseases rather than one distinct entity. As such, there has been much published on the classification of fungal rhinosinusitis (FRS) [5,6,7,8]. Early classifications of FRS used the causative organism as the descriptor, i.e., aspergillosis, mucormycosis, etc. [1]. The shift from the causative organism to the pathology of the disease process occurred in 1965, when Hora described two broad categories. These are invasive or non-invasive, dependent on the potential of the fungal hyphae to invade the tissues through the epithelium (invasive) in comparison to the infection being confined to the superficial epithelium (non-invasive) [5,7]. As its name suggests, invasive FRS can result in dramatic tissue invasion through mucosa, bone, neurovascular structures and surrounding organs [3]. As well as this distinction in the pathology of FRS, there is some variability in chronicity of the disease. This has resulted in further subdivision of FRS—acute (less than four weeks) or chronic (greater than four weeks) [5].

With these subdivisions highlighted, FRS is further categorised into six main subgroups:

### 1.3. Non-Invasive Fungal Rhinosinusitis

Saprophytic fungal infestationFungal BallAllergic fungal rhinosinusitis

### 1.4. Invasive Fungal Rhinosinusitis

Acute invasive fungal rhinosinusitisChronic invasive fungal rhinosinusitisChronic Granulomatous invasive fungal rhinosinusitis

We shall discuss the pathophysiology, diagnosis, management and current controversies of each of these in turn.

## 2. Non-Invasive Fungal Rhinosinusitis

### 2.1. Saprophytic Fungal Infestation

Saprophytic fungal infestation is traditionally described as fungal colonization of the secretions of the sinonasal cavity or crusted mucosa. As such, it was not always included as part of the classification of FRS. This usually follows surgical intervention leaving an inflamed or crusted nasal mucosa, which then gets infected with inhaled fungal spores [9]. It has no invasive features and is confined to the crusts/mucosa within the nasal cavity.

Diagnosis:

This can be an asymptomatic disease process and so often goes undiagnosed. It can, however, present with a foul smell in the nasal cavity [3].

Management:

The main relevance of this category of FRS is the speculation that it can be the starting point for fungal ball development [10]. There is, however, a paucity of literature on this category of FRS. If symptomatic, generally, this is managed as part of concurrent sinus surgery with the clearance of the crust, or more commonly, non-surgically, with nasal douching; it does not require formal surgical intervention [3].

### 2.2. Fungal Ball

Fungal balls are defined as a densely matted collection of fungal hyphae, which exist extra-mucosally, causing minimal mucosal inflammation or reaction [3,8]. Previously, these have been termed *‘Aspergillomas’*, owing to the most commonly encountered fungus being *Aspergillus* species [5]. However, other fungi have been implicated in fungal ball production and hence the change in terminology to ‘Fungal ball’ [3,5]. These fungal balls frequently occur in only one sinus with the most commonly affected being the maxillary sinus (94%). The majority of the remaining cases occurring in the sphenoid [3]. In contrast to invasive fungal sinusitis, fungal balls typically occur in immunocompetent patients and are reported to be more prevalent in the middle age female population [11,12]. The pathogenesis of fungal ball formation is not completely understood. As with saprophytic fungal infestation it is widely accepted that the mechanism of fungal introduction to the sinus is via inhaled spores [3]. As previously mentioned, it is possible that previous surgery/mucosal injury might have a role to play [5]. In addition to this, there is an association of maxillary fungal balls with previous dental treatment, more specifically dental fillings and iatrogenic oroantral communication [12,13,14]. This is because certain components of sealers used for endodontal treatment (for example Zinc Oxide) may promote fungal growth. This can, however, take years to manifest after the initial intervention [3]. The association has been shown to be quite strong, with studies reporting rates as high as 89.2% in patients who have had previous dental work [15]. As such, it has been suggested that panoramic dental imaging should be a mandatory part of the management of patients with suspected fungal balls [13]. Anatomical variation and obstruction at the osteomeatal complex is thought to be a causative factor in fungal ball formation, but this is somewhat debated in the literature [5,12,13]. It is worth noting that fungal balls might occur alongside other forms of fungal sinus disease [3].

Diagnosis:

Presentation of a fungal ball is usually non-specific and indeed it can be asymptomatic. As such, this is often encountered as a part of investigation and treatment for chronic rhinosinusitis (CRS) [12]. Some symptoms that cross over between the two disease processes include facial pain (or retro-orbital pain in sphenoid disease), post nasal drip and cacosmia [3]. Endoscopic examination of the nasal cavity produces a variety of different findings in the setting of fungal balls. This may range from an entirely normal mucosa and nasal cavity, through to crusting, purulent discharge and oedematous mucosa with polyp formation [5,12]. This makes it difficult to clinically distinguish this entity from conventional CRS. CT scans of the paranasal sinuses are commonly performed, whereupon the radiological suspicion of a fungal infection or fungal ball can be raised. The sensitivity and specificity of this modality for diagnosing fungal balls are 62% and 99%, respectively [14]. This is borne by intralesional hyperdensity or calcification within an opacified sinus [12]. The CT will also allow the evaluation of anatomical variations, the identification of foreign bodies (including dental amalgam) and any associated dental pathology.

Further suspicion toward a diagnosis of fungal ball can be made intraoperatively. The classical signs on nasoendoscopy are: ‘cheesy’ and ‘clay-like’ inspissated mucous. This is highly sensitive (100%) and specific (99%) [12].

Fungal balls are diagnosed histologically. The classical findings are of matted fungal hyphae, separate from the sinus mucosa. There are no elements of invasion or granulomatous change in surrounding mucosa, bone or blood vessels, when examined histopathologically [3]. In fact, Deshazo proposed a definition of fungal balls as having the specific histopathological findings (a dense matt of fungal hyphae) as described above with no invasive features [16]. *Aspergillus* species are the most frequently encountered organism (reportedly over 90%), but this is mostly a suspicion from the classical histological appearances of *Aspergillus*, as fungal culture is required to prove this [12]. Microbiological cultures are somewhat unreliable when it comes to fungal balls. The positive rates of fungal culture are frequently reported to be low (32.1%) [12]. This has been linked to the poor viability of the fungal hyphae [12].

Management:

As fungal balls are not invasive, systemic or topical, medial therapy with antifungals are not appropriate. As such, this disease is primarily managed by functional endoscopic sinus surgery (FESS). The management should be targeted at the affected sinus and any contributing factors (i.e., oroantral fistula or retained dental amalgam). The fungal material should be macroscopically cleared and the sinus washed out. It is also important to sample surrounding mucosa, to rule out invasive fungal sinus disease [3]. However, other approaches, such as an osteoplastic approach (operating endoscopically via a window in the anterior wall of the maxillary sinus) have been described in the literature [13].

### 2.3. Allergic Fungal Rhinosinusitis

The third form of non-invasive fungal sinusitis is allergic fungal rhinosinusitis (AFRS). This is thought to be the most common form of fungal sinus disease [1]. It has a similar presentation and set of findings to CRS and, as such, the existence of this condition has been discussed in the literature and its isolation from CRS has been questioned. It has, however, some specific features which have been included in diagnostic criteria. These are based upon the unique set of clinical, radiological and histological findings [17]. The first recognition of this condition occurred in the 1970s as a result of the similarities in the mucous found in AFRS and allergic bronchopulmonary aspergillosis (ABPA) [3]. The most widely used diagnostic criteria was described by Bent and Kuhn (Table 1) [18]. There has, however, since been some debate about the value of this system, as some of the features described are not unique to AFRS.

AFRS typically is seen in younger patients (21–33 years) who are not immunocompromised [3]. It is thought to be more common in the atopic population [1].

It is widely appreciated that the exact pathological processes surrounding AFRS are not fully understood [1,3,17]. The most commonly cited pathogenic mechanisms are drawn from the similarities to ABPA. This highlights the role of type I and type III hypersensitivity reactions to fungal antigens [17]. This has been supported in the literature by Manning and Holman [19]. This means that this type of fungal sinus disease is not a true fungal infection, rather, a reaction to fungal antigens. A type I hypersensitivity reaction is an IgE mediated reaction to an antigen. This results in the release of histamine and inflammatory mediators resulting in an eosinophilia. This has been shown to be the case on mucosal biopsies of patients with AFRS [3,19]. A type III hypersensitivity reaction occurs as a result of an IgG mediated antigen-antibody complex formation. This results in immune complex deposition and inflammation. It is thought that, via these two mechanisms, inhaled fungi, causing a mucosal reaction resulting in inflammation and importantly mucous production, termed ‘eosinophilic’ mucin. This is a thick and dark mucous secretion. This mucin classically shows eosinophilic inflammatory cells and ‘Charcot–Leyden’ crystals. These crystals are the breakdown products of cells by eosinophilic enzymes [19,20]. There has been some debate about the validity of these theories as ‘allergic mucin’ has been shown to be present in patients with non-allergic CRS, albeit in small volumes [3]. This may be explained by evidence that some patients have an abnormal reaction to fungi, without necessarily fulfilling the criteria for AFRS. Therefore, it is suggested that fungi may cause CRS, but that this is separate from classical AFRS [3]. Adding to this debate are studies that have reported high levels of serum IgE and IgG in patients with AFRS compared with CRS with nasal polyps (IgE often over 1000 IUmL). However, there are studies that state no significant difference [21,22]. Inflammatory mediators have been shown to have an important role in AFRS (e.g., T-Helper cells, cytokines, interleukin-4 and interleukin-5). These mediators have been shown to have higher levels in the setting of AFRS when compared with healthy individuals [22]. For these reasons, the pathogenesis of AFRS is complex, controversial and incompletely understood [21].

Diagnosis:

AFRS is often suspected in the setting of a patient who is resistant to the routine treatments, including surgical intervention, for CRS. They may, however, show good, but relapsing, responses to oral steroids [17]. In addition to fungal mucin production, another strongly associated feature of AFRS is the production of nasal casts. These are dark coloured, firm or rubbery casts of fungal mucin [17]. Although the disease is non-invasive in nature, it is not uncommon for these patients to show evidence of bony erosion on cross sectional imaging and to have ophthalmological findings e.g., proptosis [20]. The most commonly affected sinuses are the ethmoids, which may account for this finding [17]. The disease is often asymmetrical.

As with most sinus pathology, the imaging modality of choice is CT scanning. The classical findings of AFRS on CT are the ‘double density’ sign. This is due to thick fungal mucin (which may contain heavy metal deposits) surrounded by hyperplastic mucosa. As mentioned above, there may be a suggestion of bony erosion or expansion without any signs of invasion [3]. Both T1 and T2 weighted MRI scans, in combination with CT, can help add weight to the suspicion of AFRS. Both T1 and T2 weighted scans show peripheral enhancement. T1 scans often show hypo- or iso-intense sinus contents, while T2 scans can easily be misinterpreted as normally aerated sinuses due to protein content of the fungal mucin [3].

As mentioned earlier, fungal culture is difficult to rely upon; firstly, fungi are nearly ubiquitous in the sinuses of healthy patients; secondly, negative fungal cultures do not rule out the presence of fungal matter [3].

To try and create a uniform diagnosis for AFRS, several classification systems have been suggested. The most widely accepted is the one proposed by Bent and Kuhn [18]. This system splits diagnostic criteria into ‘major’ and ‘minor’ criteria. All of the major criteria should be fulfilled to confirm a diagnosis of AFRS, with minor criteria used to support the diagnosis. Other classifications exist, which exclude the need for atopy to be part of the diagnostic criteria, to account for the non-atopic population [3]. There have been very few studies that try to identify a useful diagnostic tool to help diagnose this condition with any more accuracy than these classifications [22]. A recent prospective study looked at multiple preoperative markers, to attempt to identify a method for preoperative diagnosis. The authors concluded that the only marker that could confirm a diagnosis of AFRS was a total IgE of over 517 kIU/L [22]. They also suggested that a combination of MRI and CT improved the accuracy of preoperative radiological diagnosis [23].

Management:

There are roles for both medical and surgical treatments in patients with AFRS. However, patients have often undergone several medical and surgical treatments prior to a diagnosis of AFRS. In fact, according to Bent and Kuhn, mucin and fungal stains are required in order to obtain a diagnosis. These are most commonly obtained at the time of surgical intervention. As such, most of the evidence on management of AFRS comes in the form of post-operative medical treatment [1,3,21,24].

It is generally appreciated that, due to the anatomical distortion of sinus outflow tracts, association with orbital involvement and the high rate of early recurrence in non-surgically treated patients, surgery in the form of full clearance of the fungal material and mucin as well as restoration of sinus drainage pathways is key [20,21,22]. The need for more aggressive clearance and surgery in AFRS highlights the need for an accurate preoperative diagnosis. It is important that the correct samples be sent for histological and microbiological examination at the time of surgery. This includes samples of polyps and the mucin being sent for histology, fungal staining and culture [3]. The presence or absence of invasion is a key factor to be highlighted in the histology.

The mainstay of post-operative medical treatment has been outlined by Gan et al. in 2014. This recommends the routine use of oral and topical steroid treatment. The use of oral antifungals and immunotherapy is reserved for refractory cases [24].

Steroid therapy:

The use of corticosteroids (oral and topical) is widely shown to be beneficial in AFRS for the same reasons as in CRS. Suppression of inflammatory responses, eosinophilia and IgE levels has been shown [22,24]. An exact regimen for oral steroids has not been reported; short bursts compared with a prolonged course with tapering doses have been described [3,21,22,24]. Understandably, there is a reluctance for prolonged oral therapy for fear of provoking adverse side effects [21]. The evidence for the use of topical nasal steroids alone is limited in AFRS. There is evidence supporting its use combined with the use of oral steroids; rates of recurrence at two years are lower (15%) when compared with placebo (50%) [21]. The same principles of their mechanism of action and the wealth of evidence for their use in CRS with nasal polyps adds weight to their use in AFRS [21,22].

Antifungal therapy:

Oral antifungals have been used in the post-operative management of AFRS. While AFRS is not a fungal infection, the aim is to reduce the fungal load and as such reduce the immune response to it. However, their benefit is debatable. A Cochrane review on the efficacy of antifungal therapy in CRS with nasal polyps failed to demonstrate any benefit [21]. Gan et. al., however, concluded that the use of Itraconazole was beneficial in some patients and reduced the need for steroids [24]. Given the side effects of systemic antifungals along with the mixed evidence in support, they should be as a last resort in patients not adequately responding to steroid therapy [24]. The evidence for the use of topical antifungals is sparse and no recommendations have been made for their routine use [24].

Immunotherapy:

Due to the immunological theories of pathogenesis of AFRS, namely it being described as a type I hypersensitivity reaction, studies have been developed to assess its treatability with immunotherapy [22]. This has been described as subcutaneous immunotherapy, aimed at limiting the reaction to exposed fungal antigens, much the same as immunotherapy for grass pollen in allergic rhinosinusitis [21]. Systematic reviews of the literature have found evidence (grade C) supporting its use in AFRS [24]. This evidence is somewhat limited, but there is symptomatic benefit in the short term (3–4 years); long-term benefits are not well described [22]. Other than symptomatic benefits, there is evidence to support its use in reducing the need for long-term post-operative steroid use [22]. Due to the limited evidence, long courses of treatment and expenses involved, immunotherapy is reserved for those patients’ refractory to first-line treatments described above [24].

Leukotriene modulators:

The use of leukotriene modulators (i.e., Montelukast) has been investigated in the setting of AFRS with one case study suggesting that it could be beneficial in AFRS refractory to conventional therapy. However, additional studies are needed [3,21,24].

Biological therapies:

More recently, the investigation of biological therapies aimed at suppressing inflammatory mediators (e.g., anti-IgE and anti-IL-5 agents) have been a suggested treatment. This has been extrapolated from their use in asthma. Early studies focusing on the use of biological therapies (e.g., omalizumab and mepolizumab) have shown promising results [22,25,26]. There were improvements in SNOT-22 scores (31%) as well as endoscopic scores (61%) in a patient population treated with omalizumab for AFRS and asthma [22]. However, large scale prospective randomized control trials aimed at their use in AFRS are required to investigate their benefit before recommendations can be made on their use [22].

## 3. Invasive Fungal Rhinosinusitis

### 3.1. Acute Invasive Fungal Rhinosinusitis

Acute invasive fungal rhinosinusitis (AIFR), although rare, is important because of its aggressive course and high mortality rates (around 50% but with some reports of up to 80%) [27,28]. From the available literature, this appears not to have significantly changed over the last 20 years [28]. As the name suggests, AIFR differs from non-invasive forms of fungal rhinosinusitis in that there is invasion of primarily neural and vascular structures (rather than mere mucosal colonization). Typically, the fungal spores are inhaled, following which (owing to disruption of the normal immunological responses) the fungi grow on the mucosal lining and invade neurovascular structures [3]. This, in turn, causes thrombosis with local and/or distant ischemia leading to necrosis. This facilitates spread outside the infected sinus cavity into surrounding tissues and (commonly) the bone. A definition of AIFR has been proposed as ‘the presence of fungal hyphae within the sinonasal mucosa, submucosa, vasculature or bone, in the setting of one month or less of sinusitis symptoms’ [28,29].

There are two common causative organisms of AIFR; these are typically from the *Aspergillus* species and Zygomycetes order. As mentioned previously, this gave rise to the well-known but outdated term of Mucormycoses (owing to the frequent isolation of Mucor).

In contrast to the non-invasive forms of fungal rhinosinusitis, AIFR is most commonly encountered in those patients with a form of immunocompromise. These broadly fall into two categories and each of these have commonly associated pathogens with them. The first are diabetic patients (roughly 50%), particularly if poorly controlled, and is frequently associated with diabetic ketoacidosis. This subset of patients frequently have Zygomyecetes order isolated [3]. This is due to their affinity for acidotic environments with high glucose concentrations. The second subset of patients are those who are immunosuppressed—for example, with neutropenia, HIV/AIDS, hematological malignancies and patients receiving chemotherapy [27]. Although neutropenia is strongly associated with AIFR, the vast majority of these patients have a hematological malignancy [29,30]. These patients often have *Aspergillus* species isolated. There also appear to be an additional small subset of patients with a propensity to develop AIFR. These are those who are iron overloaded or in renal failure and receiving deferoxamine for iron chelation. The mechanism behind this is that some fungi (*Rhizopus*) can bind to deferoxamine which in turn supplies the fungus with extra iron which aids its growth [31].

There is evidence to suggest that diabetic patient’s survival is better than those with immunosuppression [3]. This has been attributed to the more easily optimised disease state. However, there is no evidence to suggest that the specific pathogen isolated can help inform prognosis [28,30]. Other factors which appear to be related to a poorer prognosis include: older patients; aplastic anaemia; delayed diagnosis; concurrent hepatorenal failure; intracranial complications and neutropenia [28,30].

Diagnosis:

Early diagnosis and initiation of treatment is paramount to improving survival in AIFR [29]. One difficulty with this is that the initial prodrome may be relatively innocuous and nonspecific. The presentation may be with rhinorrhea (often clear), nasal congestion and facial pain or pressure and fever [28]. However, it has been noted that even in advanced AIFR there may be only very few or even no classical symptoms associated with rhinosinusitis [1,3,29]. This non-specific course is for four weeks or less (as per the definition of AIFR). Adding to the difficulty of timely diagnosis, the ensuing invasive symptoms may present rapidly and worsen over a matter of hours [3]. These symptoms are often dictated by invasion into local anatomy. These can consist of numbness, facial swelling and erythema, proptosis, diplopia, visual loss, headaches and neurological deficit. Nerve deficits (cranial nerves III, IV and VI) can indicate the involvement of cavernous sinus [3].

Examination findings depend upon the stage of the disease at presentation. In the early stages, there may be subtle mucosal changes, such as a pale or odematous nasal mucosa [28]. There may be some associated early signs of neural invasion such as anaesthesia [3]. As the disease progresses and neurovascular invasion worsens the pale mucosa becomes more avascular and turns darker and eventually black and necrotic. This mucosa will then ulcerate and slough off, forming a thick eschar or crust [3,28]. As well as changes within the sinus cavities, similar findings can be found on the nasal septum and can cause septal perforations [3].

In case of suspicion of AIFR, a panel of blood tests should initially be requested—full blood count to assess for neutropenia or signs of hematological malignancies, renal function, blood glucose, capillary ketones, iron level and tests for infectious diseases such as HIV infection [28]. These will help identify a cause for the susceptibility to AIFR to enable a more targeted therapy.

With regard to imaging, as with other forms of fungal rhinosinusitis, the first imaging modality is a CT scan. The findings are not specific in the first instance. They are typically comprised of mucosal thickening in an asymmetrical fashion. Complete absence of mucosal thickening is very sensitive for the absence of AIFR [28]. In more advanced disease, the scan will reflect the further local invasion and bone erosion. Invasion of the fat surrounding the maxillary antrum has been shown to be a highly specific finding for AIFR infecting the maxillary sinus [28]. Further imaging, with a contrast enhanced MRI, is often required. This has been shown to be more sensitive than CT at identifying AIFR [28]. MRI allows better evaluation of the extent of spread outside the sinus (i.e., orbit, brain, soft tissues). It also allows earlier detection with subtle changes in the mucosa (loss of enhancement), fat and extraocular muscles [28]. In addition to this, MRI has been shown to have a role in prognostication in AIFR. Specific contrast enhancement patterns have been described in AIFR. Lack of contrast enhancement has been shown to be of poor prognostic value [32]. The areas of lack of contrast enhancement correspond with the presence of high fungal load with areas of coagulation necrosis [32].

To confirm the diagnosis of AIFR, a tissue biopsy is required. The evidence suggests the most frequently affected sites, and hence the most sensitive for biopsy, are the middle turbinate (75–86% sensitivity and 100% specificity), the nasal septum and floor of the nasal cavity [28,33]. The most frequently affected sinuses are the maxillary and ethmoid sinuses. An improvement in mortality was shown by one study that had a low threshold to biopsy the middle turbinate in the presence of neutropenia, mucosal pallor and mucosal inflammation on CT [33]. The sample should be processed for histopathology and culture (although the sensitivity of culture is again very low) [3]. The frozen section has been shown to be a sensitive (84%) and specific (100%) method of establishing a diagnosis in a timely fashion [3,34].

Management:

The management of AIFR has three arms: reversal of pre-disposing state (i.e., neutropenia, ketoacidosis, etc.); surgical debridement and antifungal therapy. As such, a multidisciplinary approach is key to successful management. Early involvement of the appropriate medical (e.g., diabetes, hematology, etc.) and surgical (ENT, maxillofacial, neurosurgery and ophthalmology) specialties along with seeking microbiology advise is fundamental to successful treatment [3].

Surgical intervention is well recognised as a crucial element of management [33,35,36,37]. This primarily takes the form of endoscopic sinus surgery with the aim of firstly establishing an early diagnosis and obtaining tissue samples [30]. The other aims of surgical intervention are debridement of necrotic tissue (mucosa and bone if necessary) [28]. It is well established that the debridement of necrotic tissue in this setting should be undertaken until healthy, bleeding tissue is encountered. This may well include removal of large amounts of nasal mucosa, turbinates as well as extended sinonasal access procedures to clear the sinus (e.g., medial maxillectomy, DRAF III, etc.) [28,33,35]. Added to these, quite often, repeat procedures are needed to clear any progressively developing necrotic tissue [37]. Complete surgical debridement improves survival. The use of intraoperative frozen section has a role to play in ensuring complete resection. However, it has been shown to have a 70% negative predictive value and as such needs to be combined with other indicators like a freely bleeding surface [28,34]. Clearly, in patients with more advanced AIFR involving the orbit or intracranial complications, the question arises as to whether a more extensive and disfiguring open surgery is required. While the evidence suggests that procedures such as orbital exenteration or radical maxillectomies do not significantly improve survival, it must be noted that these patients have extremely poor survival anyway [3,30]. Many authors agree that orbital exenteration should only be performed in the case of a non-functioning eye [38]. Another point of controversy pertains to the timing of surgery—should this be managed as a medical emergency, for example overnight, or can surgical debridement be briefly delayed until optimal resources are available. This question is somewhat incompletely answered in the literature. The consensus is that, whilst this is an urgent procedure, definitive management can be deferred up to 24 h, should it be felt that it was to improve overall results [28].

Early instigation of systemic antifungal therapy has been shown to improve survival [28]. Obtaining appropriate fungal cultures are important to ensure that correct antifungal agents are used. The mainstay of antifungal therapy is Amphotericin B, as it acts well against Mucorales and *Aspergillus* [3]. It does, however, have a significant side effect profile, in particularly being nephrotoxic. The development of liposomal Amphotericin has reduced the nephrotoxicity to some degree [28]. Once cultures are available, Voriconazole becomes an option in patients with *Aspergillus*. It has a better side effect profile but is less effective in Mucorales [28]. Other antifungals exist (Posaconazole, Isavuconazole) but either are reserved for second line treatments and require a broader evidence-base [28], as such close working with microbiologists is fundamental to correct management.

Some evidence is available in the literature for the use of hyperbaric oxygen therapy (owing to the release of oxygen free radicals) as an adjunct to management. The evidence is limited, but appears to be of more beneficial in diabetic patients [28].

### 3.2. Chronic Invasive Fungal Rhinosinusitis

Chronic invasive fungal rhinosinusitis (CIFR) is pathologically very similar to AIFR, but occurs over a much more chronic path (months to years). This is in part due to its occurrence more frequently in the immunocompetent population [3]. Due to a slower progression, it presents more insidiously and as such can present with bloody nasal discharge, unilateral nasal obstruction, cacosmia or as a mass in the nasal cavity or paranasal sinuses that can result in proptosis. Other areas for invasion are through the maxillary sinus, causing skin changes, or into the anterior cranial fossa, causing neurological symptoms [3]. Quite often, these can be mistaken for a malignancy [39]. Similar to AIFR, *Aspergillus* and Mucor are the commonly found organisms.

Diagnosis:

As with AIFR, imaging (contrast enhanced CT scan) has an important role in diagnosis. Signs include bony destruction and hyperattenuation [3]. As mentioned above, this often leads to the initial mis-diagnosis of a malignancy. MRI with contrast can provide information on deeper invasion, meningeal enhancement or cavernous sinus involvement [3].

Early tissue diagnosis is once again important. This shows evidence of invasion of neurovascular structures. It differs from AIFR as it has a lower inflammatory response and as such will have a less dense population of inflammatory cells [3].

Management:

The management is broadly the same as AIFR, with antifungals and surgical debridement. The principals of surgical resection are the same, as are the approaches to antifungal choice [3]. Clearly, reversal of the immunocompromise state is not a factor in management as CIFR is predominantly encountered in immunocompetent individuals.

### 3.3. Chronic Granulomatous Invasive Fungal Sinusitis

Chronic granulomatous invasive fungal sinusitis (CGIFS) is infrequently seen in the western world. It exists more commonly in Northern Africa, the Middle East and Asia in either immunosuppressed or immunocompetent patients [3]. As with AIFR and CIFR, it involves invasion past the submuosa [3]. This is typically associated with *Aspergillus*. Its histological distinction comes from it being a disease process that forms non-caseating granulomas. It is known to occasionally co-exist with other forms of fungal sinus disease.

Diagnosis and management is the same as with other forms of invasive fungal sinusitis-surgical debridement and systemic antifungals form the crux of therapy.

## 4. Summary

The salient points for each subtype of fungal sinusitis have been summarised in Table 2, including the key features, diagnostic strategies and management options.

## 5. Conclusions

The role of Fungi is clearly important to ENT surgeons and has been subject to much discussion in the literature. The impact of fungi in the upper airways is wider than just its role in fungal sinusitis. Fungi have been postulated to have a developmental role in CRS [40]. Although this has since been disproven in the literature, it has been demonstrated that they form a regular commensal of the sinonasal tract [41]. This must be borne in mind when considering that pathophysiology of sinonasal diseases.

The broad spectrum of diseases seen in fungal sinus disease makes it an interesting and challenging disease process to understand and manage. We have outlined the spectrum of diseases encountered and the relevant management strategies informed by the recent literature. With the increasing awareness and prevalence of this spectrum of conditions, further research will help identify appropriate medical management strategies to supplement existing surgical treatments. Further evaluation of the use of biological therapies in AFRS is likely to be the biggest change in the near future.

## Figures and Tables

**Table 1 medicina-55-00319-t001:** Bent and Kuhn criteria.

Major Criteria	Minor Criteria
Type I hypersensitivity (IgE testing, skin testing or clinical history)	Unilateral disease
Nasal polyps	Asthma
Characteristic radiology (CT findings)	Bone erosion/expansion on CT
Presence of eosinophilic mucin without invasion	Fungal cultures
Positive fungal stain	Eosinophilia
	Charcot–Leyden crystals in mucin

**Table 2 medicina-55-00319-t002:** Summary table.

Subtype	Key Features	Diagnosis	Management
**Saprophytic fungal infestation**	Non-invasive Usually follows surgical intervention Frequently asymptomatic	Clinical No radiology required	Conservative–douching, surgical intervention only if required for other disease process.
**Fungal Ball**	Non-invasive Immunocompetent patients Densely matted balls of extra-mucosal fungal hyphae Most commonly affects Maxillary sinus Strong association with previous dental procedures/pathology	Endoscopic examination may range from normal mucosa through to crusting, purulent discharge and oedematous mucosa with polyps. Cheesy clay like material encountered intraoperatively. CT sinuses Consider panoramic dental imaging Histological examination of fungal matter.	Endoscopic sinus surgery and macroscopic clearance of fungal matter. Rule out invasive disease by sampling adjacent mucosa. Address any contributing factors (i.e., oroantral fistulas)
**Allergic fungal rhinosinusitis**	Most common form of fungal sinus disease. Non-invasive Younger, immunocompetent, atopic individuals Can be considered a hypersensitivity reaction to fungal antigens Association with presence of fungal mucin containing Charcot–Leyden crystals. Controversies over diagnosis and links with CRS. Consider in patients with suspected CRS resistant to conventional treatments. Evidence of bony erosion on cross sectional imaging.	CT sinuses T1 and T2 weighted MRI Bent and Kuhn criteria (Table 1) Serum IgE levels	Functional Endoscopic Sinus Surgery aimed at clearing fungal mucin and restoration of functional sinus drainage pathways. Post-operative topical and oral steroid therapy Consider oral antifungals in refractory cases Consider immunotherapy in refractory cases Further evidence needed for use of Montelukast or Biological therapies.
**Acute invasive fungal rhinosinusitis**	Invasion of neurovascular structures causing necrosis and invasion outside of sinus cavity with distant complications including ophthalmological and neurological complications Rare Previously termed ‘Mucormycosis’ Aggressive with high mortality rates (50−80%) Association with diabetes, immunocompromise and iron overload or iron replacement therapy Presentation with history of classical sinusitis symptoms for up to one month	Presence of cranial nerve, neurological or ophthalmological complications Endoscopic findings or necrotic mucosa Blood tests including assessment for causes of immunocompromise. Cross sectional imaging of sinuses and orbit +/− brain with contrast CT +/− contrast enhanced MRI scan Biopsy of nasal mucosa (most sensitive areas are middle turbinate, nasal septum and floor of nasal cavity) for histology and culture.	Reverse/optimise predisposing state/immunocompromise Surgical debridement with endoscopic sinus surgery to clear necrotic tissue and consider use of open procedures/requirement for orbital exenteration if required. Consider use of intraoperative frozen section. Early systemic antifungal therapies guided by cultures. Consider role of hyperbaric oxygen therapy in diabetic population.
**Chronic invasive fungal rhinosinusitis**	Invasive Similar to AIFR but over a more indolent path of months to years. More commonly immunocompetent patients Frequently mistaken for malignancy	Contrast cross sectional imaging as with AIFR Early mucosal biopsy as with AIFR	As with AIFR but reversal of predisposing factors is less relevant as it commonly occurs in immunocompetent individuals.
**Chronic Granulomatous Invasive Fungal Sinusitis**	Uncommon in western world–more frequently seen in North Africa, Middle East and Asia. Immunocompetent or immunocompromised patients Forms non-caseating granulomas	As in CIFR Key differentiation is the presence of non-caseating granulomas on histological examination	As in CIFR

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
