# Peer review of "Invasive and Non-Invasive Fungal Rhinosinusitis—A Review and Update of the Evidence"

_medicina, 2019, doi:10.3390/medicina55070319_

Round 1

Reviewer 1 Report

This manuscript is a well-organized, comprehensive review of the various types of fungal sinusitis. It effectively distinguishes invasive from non-invasive types and exhaustively discusses the salient concepts in diagnosis and management.

The main critique that can be put forth about this study pertains to grammatical errors that are present throughout. Significant effort should be made to revise these errors prior to consideration for publication. Otherwise, this manuscript cannot be accepted for publication in the present form.

With respect, to the subject content put forth, it is spot-on. A few minor topics which warrant consideration are detailed below:

The authors comment on the potential use of biologic agents in the treatment of AFRS, including the use of anti-IgE and anti-IL-5 agents. While IgE is discussed in the pathogenesis of this disease, there is not mention of IL-5 or other Th2 pathway mediators in the pathogenesis of this disease. Are their sources that discuss this? If so, this should be discussed.

There have been case reports about the conversion of AFRS to AIRS due to the use of systemic corticosteroids. This data should be included when discussing the pathogenesis of AIRS.

Author Response

Thank you for your review and comments on our manuscript.

We are pleased you found it informative and well organised.

We have taken the opportunity to revise the manuscript in light of your comments. We have under taken a thorough grammatical review of the paper and made relevant changes via 'track changes'.

We have also updated the section on AFRS to include more detail, within the scope of the paper, on the pathogenesis and the potential targets for biological therapies.

With regards to the reports on conversion of AFRS to AIFR, we were unable to find any robust evidence or case reports to highlight this. We have therefore not included this within the amended manuscript.

We hope you find the changes have improved the manuscript appropriately.

Yours Sincerely,

Reviewer 2 Report

Dear, 

I've read the paper invasive and non-invasive fungal rhinosinusitis – a review and update of the evidence with great interest. 

This paper is addressing a difficult and little studied area in rhinology and summarised the available studies in a very nicely structured and complete way.

I feel little can be added to this review. It might be worthwhile mentioning the studies from Ponikau suggesting a major role of fungi in all CRS patients, which was rejected by the Amphocetirizine study from Ebbens and Fokkens.

Also the paper from Choi et al. https://doi.org/10.1007/s00234-018-2034-0 linking MRI findings to prognosis in AIFR might be added.

Furthermore I have nothing to add

Author Response

Thank you for your review and comments on our manuscript.

We have taken on board your comments and added references on the topics of CRS and also MRI imaging in AIFR.

We have also taken the opportunity to correct some grammatical errors.

The changes have been made with, and are visible via track changes.

Many Thanks,

Peter